# Demographic Influences on Adult HPV Vaccination: Results from a Cross-Sectional Survey in Tennessee

**DOI:** 10.3390/healthcare12131305

**Published:** 2024-06-29

**Authors:** Alina Cernasev, Oluwafemifola Oyedeji, Cary M. Springer, Tracy M. Hagemann, Kenneth C. Hohmeier, Kristina W. Kintziger

**Affiliations:** 1Department of Clinical Pharmacy and Translational Science, College of Pharmacy, University of Tennessee Health Science Center, 301 S. Perimeter Park Dr., Suite 220, Nashville, TN 37211, USA; thageman@uthsc.edu (T.M.H.); khohmeie@uthsc.edu (K.C.H.); 2Department of Public Health, University of Tennessee, Knoxville, 390 HPER, 1914 Andy Holt Ave., Knoxville, TN 37996, USA; oonaade@vols.utk.edu; 3Research Computing Support, Office of Innovative Technologies, University of Tennessee, Knoxville, 2309 Kingston Pike, Suite 132, Knoxville, TN 37996, USA; springer@utk.edu; 4Department of Environmental, Agricultural & Occupational Health, College of Public Health, University of Nebraska Medical Center, Omaha, NE 68198, USA; kkintziger@unmc.edu

**Keywords:** HPV vaccine, USA, Southern state, survey

## Abstract

HPV is the most prevalent sexually transmitted infection in the U.S., with more than 80% of all Americans contracting it by age 45. Effective vaccines for HPV exist and were recently approved for adults aged 27–45 years, though uptake remains low in all age groups, particularly in Tennessee where 1089 cancers were attributed to HPV in 2020. Between 29 June and 17 August 2023, we conducted a cross-sectional survey to gain insights about the barriers and facilitators of HPV in 2011 adults aged 18 to 45 years in Tennessee. We developed our survey based on previous instruments to understand predictors of HPV vaccination in adults. Using descriptive statistics and bivariate and logistic regression analyses, we found higher vaccination rates in females, participants aged 18–38 years, participants with a high school education or higher, Hispanic or Latine individuals, and participants identifying as moderate or liberal. These insights highlight the need for public health interventions that consider demographic differences to successfully increase vaccination rates and reduce HPV-associated cancer risk.

## 1. Introduction

Human papillomavirus (HPV) is the leading cause of various cancers, including cervical, oropharyngeal, anal, penile, vaginal, and vulvar cancers. According to the Centers for Disease Control and Prevention (CDC), it is associated with approximately 37,000 new cancer diagnoses (14.3 cases of cancer per 100,000 adults 18 years or older) annually in the United States (U.S.) [1] Cervical cancer, with 25,689 reported cases among women (19.3 cases of cervical cancer per 100,000 female adults), and oropharyngeal cancer, with 21,022 cases among men (16.4 cases of oropharyngeal cancer per 100,000 male adults), are the most common types of cancers caused by HPV [2,3,4].

Before the approval of the first HPV vaccine, the healthcare team relied on screening as a prevention measure to combat these cancers. Since the approval of the HPV vaccine by the Food and Drug Administration (FDA) in 2006, vaccination has become the primary tool to prevent the spread of various forms of HPV-attributable cancers [5]. While there has been some screening for anal cancer among HIV-positive men who have sex with men (MSM), this is generally restricted to a few areas in a limited number of countries [6,7]. Currently, the Advisory Committee on Immunization Practices (ACIP) recommends routine HPV vaccination of both females and males at 11–12 years of age, with catch-up vaccination recommended through age 26 years [8]. Per CDC guidelines, the HPV vaccine can be initiated at the age of 9 [9]. For those 27 to 45 years old who were not vaccinated at a younger age, the vaccine might be recommended based on shared decision making [9]. The HPV vaccine is readily available and effective at preventing HPV-associated cancers; however, coverage and completion rates have been historically low despite being included as a Healthy People 2030 objective of 80% coverage of males and females ages 13–15 [10]. Thus, championing efforts to enhance routine vaccination could prevent most HPV-related cancers, given sufficient time to reach adequate coverage [11].

Although there has been an increase in HPV vaccination uptake in the last two decades, disparities continue to exist in the rates of vaccination across the U.S. Rahman et al. showed that women in the South presented with the lowest vaccination rates in the U.S. [12]. Lu et al. examined adult vaccination coverage concerning demographic characteristics and regional variations [13]. The study reported that U.S.-born individuals generally exhibited higher vaccination rates than foreign-born counterparts, with notable disparities observed across different vaccines [13]. The findings also showed that foreign-born adults were less likely to specifically receive HPV vaccination compared to their U.S.-born counterparts, thus reinforcing the need for targeted interventions to address vaccination disparities among diverse population groups [13].

Utilizing data from the 2012 Behavioral Risk Factor Surveillance System survey across eight states, Rahman et al. assessed HPV vaccine adoption among young adults aged 18–26 alongside associated obstacles [14]. Among 3727 respondents, initiation rates were 6.3% for men and 40.4% for women, with completion rates at 1.7% and 27.4%, respectively [14]. Regional disparities were evident, particularly in the South, where uptake was notably lower in both males and females [14]. The study showed that women residing in the West and South faced barriers to initiating and completing vaccination compared to those in the Northeast [14]. In a recent study, Boersma et al. found that vaccination rates for one or more doses of the HPV vaccine in adults 18–26 years old increased from 22.1% to 39.9% from 2013 to 2018 [15]. Women were more likely to receive at least one dose of the HPV vaccine compared to men [15]. However, the situation is more critical in the Southern states where vaccination rates are significantly lower compared to the North. The suboptimal HPV vaccination coverage is more alarming in Tennessee [16]. The CDC reports that 1089 cancers (20.7 cases of cancer per 100,000 adults) were attributed to HPV in Tennessee in 2020 [1].

Multiple barriers fuel vaccine hesitancy, including concerns about the cost of the vaccine, its safety, a lack of perceived benefit, lower resources allocated for rural areas, and a lack of provider recommendation [12,17]. In a qualitative study conducted with stakeholders in North Carolina and South Carolina, Fish et al. provided valuable information about the obstacles faced by rural communities and possibilities for enhancing HPV vaccination rates in rural areas [18]. The study revealed that obstacles to HPV vaccination that are unique to rural populations, including a scarcity of healthcare practitioners and restricted access to high-speed internet [18]. In a cross-sectional survey conducted in the U.S., the respondents were predominantly female, over 26 years old, White, and non-Hispanic [17]. The study reported that willingness to receive the HPV vaccine within the next six months was generally low among participants, with various barriers cited, including doubts about vaccine necessity, safety concerns, uncertainties regarding insurance coverage, and financial constraints [17]. Factors associated with negative vaccine intention included older age, female gender, non-Hispanic White ethnicity, and a lower level of education [17].

Scholars posit that a strong provider recommendation has been the most essential factor in HPV vaccination acceptance [19,20,21]. Yet, a systematic review of studies published between 2012 and 2019 identified age, geographic, socioeconomic, and racial disparities in HPV vaccination recommendation, with fewer recommendations for younger patients living in the South, non-White, low-income, or uninsured [22]. In these studies, the proportion of parents who received provider referrals to obtain the HPV vaccine varied from 24% to 88% [22]. Overall, 13 of the 52 included studies provided single estimates of recommendation prevalence without cross-group or cross-setting comparability, whereas 39 of the 52 studies showed results stratified by the adolescent demographic characteristics highlighted in this analysis [22].

Although the literature contains numerous examples of studies that focused on intervention efforts targeting pediatric and parent populations, factors influencing HPV vaccine uptake among adults residing in Tennessee have yet to be explored in depth [23,24,25,26]. Tennessee is an ideal population on which to focus vaccination efforts, because of its lower adolescent vaccination and higher HPV-associated cancer rates. ACIP recommendations allow the population to make informed health decisions. Thus, this study aimed to better understand the demographic and socioeconomic factors that are important in predicting HPV vaccination status in Tennessee residents, specifically those aged 18–45 years.

## 2. Methods

### 2.1. Study Design and Population of Interest

Between 29 June and 17 August 2023, we conducted a cross-sectional survey to understand barriers and facilitators of HPV and other recommended vaccinations in adults aged 18 to 45 years who were residents of Tennessee. We developed our survey based on previous theory-based instruments used in HPV and other vaccine studies in adults. Questions included items based on the Health Belief Model, Social Cognitive Theory, and the Theory of Planned Behavior and other questions drawn from our qualitative studies to understand vaccine-related beliefs, attitudes, and intentions in adults [27,28,29,30,31]. Our study team reviewed all relevant literature and collectively agreed on the final measures for inclusion. We first agreed on the constructs or topics that would be included in our survey. Then, we selected survey questions used in previous studies that addressed each construct. Next, we reviewed all identified questions relevant to each construct and ranked them based on clarity, conciseness, response format, item length, and contextual fit with our research questions. We selected the highest ranked items relevant to each construct, and revised wording as needed for consistency across questions and response options. The survey was reviewed by external subject matter experts and members of our population of interest for clarity and relevance. For most of our theoretical constructs, we used a 7-point Likert scale to determine positive and negative vaccine measures, with response options ranging from “completely disagree” to “completely agree”. The focus for most vaccine-related measures was the HPV vaccine; however, we also included questions related to other common vaccines (e.g., influenza, pneumococcal, COVID-19, Tdap). The survey included questions on eligibility criteria and demographic information and the status of HPV vaccination. Other questions included in the survey, but not included in this analysis were other vaccinations; general health status; general vaccine beliefs, attitudes, and barriers; and HPV-specific beliefs, attitudes, and behaviors.

This study was reviewed and approved by the University of Tennessee Health Science Center Institutional Review Board (IRB # 21-08416-XM, approved 3 November 2021) Our survey was built in and administered by Qualtrics^XM^ (Provo, UT, USA) whose processes ensured our sample represented the age, gender, race, ethnicity, religious, and residential composition of Tennessee. We included Tennessee residents between the ages of 18 and 45 years in the study. The exclusion criteria were non-Tennessee resident, over 45 years old, under 18 years old, and refusal to participate in the study. We oversampled the 18–26-year-old population to represent the group to whom the vaccine would most likely be promoted. Participants could access the self-administered, online survey from their preferred device. The survey was administered in English and included up to 72 questions, depending on specific responses. Those who had received at least one dose of the HPV vaccine vs. those who had received no doses answered slightly different versions of HPV-related questions. As this was an exploratory observational study, no formal power analyses were conducted. However, our goal was to recruit 2000 survey participants to ensure adequate representation of the population of Tennessee by age, sex, race, religion, and residence.

Age was collected as a continuous variable but analyzed as categorical variable. ZIP code of residence was categorized based on Rural–Urban Commuting Area (RUCA) Codes, based on the 2010 Census (most recent version available at time of survey and analysis) as urban core (RUCA code 1), suburban (codes 2 and 3), large rural (codes 4–6), and small rural (codes 7–10) [32]. There were 27 ZIP codes unable to be classified either because they were new ZIP codes, missing, or entered incorrectly. We collected sex as ‘sex at birth’. Other variables included race, being Hispanic or Latine (inclusive of Latino, Latina, and Latinx), education, sexual orientation, current relationship status, political leanings, and employment status. Categories for each of these variables are provided in the tables (e.g., Table 1). Some variables (i.e., race, sexual orientation, relationship status, and employment status) had to be recategorized after data collection due to small sample sizes in certain categories.

### 2.2. Statistical Methods

We used descriptive statistics, such as frequencies and percentages, to examine survey items. Next, we conducted bivariate analyses to determine associations between demographic and other characteristics and HPV vaccination status. These included Chi-square tests, with *p*-values reported, and univariable logistic regression analysis. Finally, we used multivariable logistic regression models, including variables into our full model that were significantly associated with vaccination status in bivariate analysis, to determine associations with HPV vaccination status. Backwards elimination of non-significant variables was used to obtain the final model. We used an α = 0.05 to determine significance. Odds ratios (OR) and 95% confidence intervals (CI) were reported for logistic regression results. All analyses were conducted with IBM SPSS Statistics (version 29), and significance testing was based on a Type 1 error (or alpha level) of 0.05.

## 3. Results

### 3.1. Descriptive Statistics

In total, 2011 complete surveys were received from Qualtrics. Demographic descriptive statistics are presented in Table 1. Over half (57.3%) of participants had not received an HPV vaccination. Most participants were female (52.2%), White (81.2%), and non-Hispanic (94.0%). Most participants were aged 18–28 years (49.8%), as we had oversampled this age range. The remainder of respondents were evenly split between 29–38 years and 39–45 years (each 25.1%). The most common categories of educational attainment among respondents were high school diploma (37.4%), associates or bachelor’s degrees (25.4%), and college (23.0%). Political leanings were split relatively evenly across respondents as follows: moderate (37.2%), liberal (32.2%), and conservative (30.6%). Most respondents were employed (68.0%), straight/heterosexual (77.9%), and not in a relationship (56.0%).

### 3.2. HPV Vaccination Status

HPV vaccination status differed significantly by sex, age, being Hispanic or Latine, education, sexual orientation, political leaning, and employment status (all *p*-values < 0.05; Table 2). Females were more likely to be vaccinated than males (*p* < 0.001). The proportion of respondents who were vaccinated increased with educational attainment (26.1% in those with less than a high school degree vs. 58.1% in those with a master’s, doctoral, or professional degree; *p* < 0.001). Vaccination decreased with age (50.6% in those 18–28 years old vs. 24.6% in those 39–45 years old; *p* < 0.001). Vaccination was lower for respondents who identified as heterosexual (41.4%) compared to those who identified as gay, lesbian, bisexual, queer, or asexual (47.2%; *p* = 0.029). The proportion of respondents who were vaccinated was higher for those who identified as Hispanic or Latine (62.5%) vs. non-Hispanic (41.4%; *p* < 0.001). Conservatives (36.7%) were less likely to be vaccinated than moderates (48.4%) or liberals (47.0%; *p* < 0.001). Unemployed individuals (35.4%) were less likely to be vaccinated than students (48.9%) or those who were employed (44.8%; *p* < 0.001). Vaccination rates did not differ relationship status (*p* = 0.293), race (*p* = 0.985), or location (*p* = 0.139).

In the full logistic regression model (Table 3), females were more likely to be vaccinated than males (OR: 1.42; 95% CI: 1.15–1.75). Compared to 39–45 year olds, 18–28 year olds were 3.01 (CI: 2.30–3.95) and 29–38 year olds were 2.34 (CI: 1.74–3.15) times as likely to be vaccinated. All education levels were more likely to be vaccinated than individuals without a high school degree. Hispanic or Latine individuals were 2.01 (CI: 1.30–3.09) times as likely to be vaccinated. Compared to conservatives, moderates and liberals were more likely to be vaccinated. Sexual orientation and employment status were not significant factors in the full model. Using backwards elimination to simplify the model, only sexual orientation was removed as non-significant. The final model (Table 4) demonstrated similar associations between potential predictors and HPV vaccination status as the full model, with employment status becoming significant. Specifically, those who were employed were 1.32 times as likely to be vaccinated compared to unemployed individuals (CI: 1.02–1.70).

## 4. Discussion

Considering the suboptimal HPV vaccine coverage statistics and noting that most hesitancy research has been conducted among children and adolescents, this study aimed to assess barriers to HPV vaccination in Tennessee residents, specifically those aged 18–45 years. Previous research examining HPV vaccine barriers had a national focus among 18–26 year olds, making our study the first to investigate HPV vaccine uptake barriers in Tennessee’s representative adult population of men and women aged 18–45.

We found that females were more likely to be vaccinated (OR: 1.42; 95% CI: 1.15–1.75). Our findings are similar to those of other studies that reported that male respondents were 81% less likely to have initiated the HPV vaccination series than females [33]. Another study showed that women were over three times as likely to initiate the vaccine [34]. Therefore, more concerted efforts are needed to enhance HPV vaccine uptake and awareness in both genders, especially in Southern states.

Our findings showed that rates of vaccination increased with education, such as college degree or higher, and decreased with age. Previous U.S.-based studies have reported that the awareness of the HPV vaccine was higher in respondents with a college degree compared to those with a high school degree [35]. Furthermore, a recently published study conducted with adults aged 27–45 showed a low rate of HPV vaccination initiation, with approximately 16% of study participants receiving the vaccine, regardless of race or ethnicity [34].

Our results have significant implications for future public health interventions targeting individuals without college degrees who are near the end of the approved age range. Thus, it is imperative to develop public health interventions to target this age group. Furthermore, healthcare providers, including pharmacists, are in an ideal position to advocate and recommend the HPV vaccine to eligible individuals in the 27–45 age range.

According to our findings, HPV vaccination rates are notably higher among females, individuals aged 18–38, those with at least some college education, moderates and liberals, Hispanic or Latine individuals, and employed respondents, among adult residents in Tennessee. Frietze et al. investigated HPV vaccine acceptance and uptake in a Hispanic community residing in the South and reported a significant association between HPV vaccine uptake, gender identity, and race [36]. The study demonstrated that approximately 65% of respondents had not received the HPV vaccine. Among vaccine recipients, approximately 10% completed the entire series [36]. Public health interventions must carefully consider these demographic differences in order to enhance vaccine coverage and reduce the risk of HPV-associated cancers. Future research should examine how these characteristics influence the facilitators and barriers to HPV vaccination.

Unemployed individuals were less likely to be vaccinated than students or employed individuals. Our findings are not surprising since most young adult students have access to health insurance and may, in fact, take greater responsibility for their own health-related decisions, irrespective of parental skepticism or biases. It is plausible to speculate that these individuals might lack health insurance due to unemployment, which means they might not have access to a healthcare provider, limiting their access to the vaccine. It is essential to develop public health initiatives that address the needs of unemployed individuals who may lack insurance. Whenever healthcare providers make these recommendations, they must be cognizant of these obstacles.

Compared to conservatives, we found that moderates and liberals were more likely to be vaccinated. A recent study determined the association between HPV vaccination awareness and political ideology among adults over 18 [37]. The study encompassed 3418 adults with an even distribution between genders, with most respondents indicating that they were White [37]. Additionally, the respondents were evenly distributed politically, with one-third identifying as moderate, one-third as liberal, and one-third as conservative [37]. The study results reported self-identified liberal respondents had higher odds of HPV and HPV vaccine awareness compared to self-identified conservative respondents [37]. Similarly, a study conducted by Saulsberry et al. among adults aged 18–59 years found that respondents who self-identified as more liberal in political ideology reported greater support for receiving the HPV vaccine [38]. The authors concluded that self-identified conservative participants are less likely to support vaccine requirements than participants who self-identified as liberal [38].

### Strength and Limitations

Self-reported data for HPV vaccination limited this analysis. Furthermore, our study included a convenience sample and was conducted over two months, which might limit its generalizability. Despite these limitations, the study also has notable strengths. First, this study uses a large sample of respondents in a Southern state. Second, the sampling strategy used by Qualtrics^®^ was developed to match the demographic characteristics of Tennessee residents, and our results indicate that this was successful. Finally, since the FDA approval for adults 27–45 years of age to receive the HPV vaccine, limited research has aimed to understand characteristics associated with HPV vaccination in this population. Even fewer studies have incorporated men and women, different races, and higher education levels in analyses. Thus, this is the first study in Tennessee to include these demographics.

## 5. Conclusions

Our results indicate that HPV vaccination status among Tennessee adult residents was higher among females, 18–38 year olds, those with at least some college or more education, those with moderate and liberal political leanings, Hispanic or Latine individuals, and those who were employed at the time of the survey than other demographic characteristics. Public health interventions should take into consideration demographic characteristic differences in order to successfully increase vaccination rates and reduce HPV-associated cancer risk. Future research should consider how these demographic characteristics impact facilitators and barriers to HPV vaccination.

## Figures and Tables

**Table 1 healthcare-12-01305-t001:** Demographic characteristics of survey respondents (n = 2011).

		Count	Column N %
Vaccination Status	Unvaccinated	1153	57.3%
	Vaccinated	858	42.7%
Location	Large Rural	327	16.5%
	Small Town/Rural	230	11.6%
	Suburban	215	10.8%
	Urban Core	1212	61.1%
Sex	Male	961	47.8%
	Female	1050	52.2%
Age	18–28 years old	1002	49.8%
	29–38 years old	505	25.1%
	39–45 years old	504	25.1%
Race	White	1554	81.2%
	Black or African American	360	18.8%
Hispanic or Latine	Yes	120	6.0%
	No	1891	94.0%
Education	Less than high school	119	5.9%
	High school graduate	752	37.4%
	Some college	462	23.0%
	Associates or Bachelor’s	511	25.4%
	Master’s, Doctoral, or Professional	167	8.3%
Heterosexual/Straight	Yes	1566	77.9%
	No	445	22.1%
Relationship	No	1111	56.0%
	Yes	873	44.0%
Political Leanings	Liberal	558	32.2%
	Moderate	644	37.2%
	Conservative	529	30.6%
Employment	Employed	1352	68.0%
	Unemployed	497	25.0%
	Student	139	7.0%

**Table 2 healthcare-12-01305-t002:** Vaccination status by demographic characteristics.

		Unvaccinated		Vaccinated		Chi-Square *p*-Value
		N	%	N	%	
Location	Large Rural	194	63.6%	133	36.4%	0.139
	Small Town/Rural	140	51.6%	90	48.4%	
	Suburban	134	59.3%	81	40.7%	
	Urban Core	673	60.9%	539	39.1%	
Sex	Male	611	62.3%	350	37.7%	<0.001
	Female	542	55.5%	508	44.5%	
Age	18–28 years old	495	49.4%	507	50.6%	<0.001
	29–38 years old	278	55.0%	227	45.0%	
	39–45 years old	380	75.4%	124	24.6%	
Race	White	897	57.7%	657	42.3%	0.985
	Black or African American	208	57.8%	152	42.2%	
Hispanic or Latine	Yes	45	37.5%	75	62.5%	<0.001
	No	1108	58.6%	783	41.4%	
Education	Less than high school	88	73.9%	31	26.1%	<0.001
	High school graduate	456	60.6%	296	39.4%	
	Some college	262	56.7%	200	43.3%	
	Associates or Bachelor’s	277	54.2%	234	45.8%	
	Master’s, Doctoral, or Professional	70	41.9%	97	58.1%	
Heterosexual/ Straight	Yes	918	58.6%	648	41.4%	0.029
	No	235	52.8%	210	47.2%	
Relationship	No	651	58.6%	460	41.4%	0.293
	Yes	491	56.2%	382	43.8%	
Political Leanings	Liberal	296	53.0%	262	47.0%	<0.001
	Moderate	332	51.6%	312	48.4%	
	Conservative	335	63.3%	194	36.7%	
Employment	Employed	746	55.2%	606	44.8%	<0.001
	Unemployed	321	64.6%	176	35.4%	
	Student	71	51.1%	68	48.9%	

**Table 3 healthcare-12-01305-t003:** Predictors of HPV vaccination status, full model.

	OR	95% CI	
		Lower	Upper
Sex (ref: Male)	1.417	1.151	1.745
Age (ref: 39–45 years old)			
18–28 years old	3.012	2.299	3.947
29–38 years old	2.341	1.742	3.146
Education (ref: Less than high school)			
High school graduate	1.960	1.161	3.310
Some college	2.054	1.197	3.523
Associates or Bachelor’s	2.324	1.358	3.976
Master’s, Doctoral, or Professional	4.691	2.542	8.654
Hispanic or Latine (ref: No)	2.006	1.304	3.086
Political Leanings (ref: Conservative)			
Liberal	1.301	1.001	1.692
Moderate	1.590	1.239	2.041
Heterosexual/Straight (ref: No)	0.949	0.737	1.223
Employment (ref: Unemployed)			
Employed	1.320	1.023	1.703
Student	1.451	0.936	2.249

**Table 4 healthcare-12-01305-t004:** Predictors of HPV vaccination status, final model.

	OR	95% CI	
		Lower	Upper
Sex (ref: Male)	1.426	1.160	1.752
Age (ref: 39–45 years old)			
18–28 years old	3.031	2.317	3.965
29–38 years old	2.350	1.749	3.156
Education (ref: Less than high school)			
High school graduate	1.949	1.155	3.289
Some college	2.042	1.191	3.500
Associates or Bachelor’s	2.305	1.349	3.937
Master’s, Doctoral, or Professional	4.645	2.523	8.554
Hispanic or Latine (ref: No)	2.011	1.308	3.094
Political Leanings (ref: Conservative)			
Liberal	1.316	1.017	1.701
Moderate	1.596	1.244	2.047
Employment (ref: Unemployed)			
Employed	1.318	1.022	1.701
Student	1.451	0.936	2.249

## Data Availability

The original contributions presented in the study are included in the article.

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
