# Peer review of "Demographic Influences on Adult HPV Vaccination: Results from a Cross-Sectional Survey in Tennessee"

_healthcare, 2024, doi:10.3390/healthcare12131305_

Round 1

Reviewer 1 Report

Comments and Suggestions for Authors

The introduction could include references to other countries in the context of HPV vaccination. One sentence at the end of the introduction is a bit too little for a description of the purpose of the study.

Material and methods: you should describe in more detail on the basis of some previous studies the questionnaire was constructed, whether it was divided, etc., and what were the criteria for inclusion in the study (possibly exclusion).

Results: The results are described too briefly. There is too much information in the tables and too little explanation of these results. Please pay attention to the p-values. 

The discussion lacks references to similar results, for example, from Europe. Please supplement. The conclusions are okay. Please provide the ethical committee approval number, or if it is not required, an explanation as to why.

Author Response

The introduction could include references to other countries in the context of HPV vaccination. One sentence at the end of the introduction is a bit too little for a description of the purpose of the study.

Response: We appreciate the reviewer’s suggestions for improving our introduction. However, we feel that our Introduction should remain U.S.-focused. The Introduction sets the stage for the current study and provides context specific to our chosen research question/objective. Our research is driven by the low uptake in HPV vaccination in the U.S.; even lower vaccination rates and poorer HPV-related outcomes in the southern U.S., and recent approval for use of the HPV vaccine in adults aged 27-45 years in the U.S. with shared clinical decision-making. We contend that a discussion of other countries in the Introduction would make this section too broad. We also respectfully disagree that one sentence is insufficient to describe the purpose of our study. A well-written purpose statement is rarely more than one sentence. However, we have edited the statement to provide additional clarity and accuracy.

Material and methods: you should describe in more detail on the basis of some previous studies the questionnaire was constructed, whether it was divided, etc., and what were the criteria for inclusion in the study (possibly exclusion).

Response: Thank you for these suggestions. We have attempted to update the section per your request related to survey development. The Methods section was further amended with more detail on the inclusion and exclusion criteria for participants.

Results: The results are described too briefly. There is too much information in the tables and too little explanation of these results. Please pay attention to the p-values. 

Response: Thank you for your recommendation. We have expanded the description of our findings in the Results section.

The discussion lacks references to similar results, for example, from Europe. Please supplement. The conclusions are okay. Please provide the ethical committee approval number, or if it is not required, an explanation as to why.

Response: We value your recommendation. The ethical approval was submitted to the journal Healthcare, but we have added a statement in the Methods addressing IRB approval. We respectfully disagree with including references from Europe and other global regions in our discussion. Recommendations for European Union (E.U.) countries by the European Centre for Disease Prevention and Control (ECDC), as well as global recommendations by the World Health Organization have very different HPV vaccination recommendations/guidelines than the U.S. There is much variability in HPV vaccination programs across Europe and globally. Several E.U. countries and most other countries external to the E.U. who have vaccination programs only vaccinate “girls” (females, aged 9-15, variable), and most countries that offer vaccination to both males and females primarily focus on the 9-15 age range (again, variable). We found a few E.U. countries that vaccinate up to age 18 and only 1 country with programs vaccinating up to age 26. We could find none mentioning shared clinical decision making in those over 26 years of age. Finally, many countries report delivering this vaccine in schools. Therefore, as our focus is on increasing HPV vaccination coverage in adults in the southeastern U.S., specific to our APIC guidelines, using traditional vaccine program delivery options available in the U.S., we felt that comparisons with other countries would not add meaningfully to our discussion.

Reviewer 2 Report

Comments and Suggestions for Authors

Type of the Paper (Article)

---- > Abstract

1.       Avoid the term "Latino/a/x individuals" is difficult to understand, and not adequate for an academic journal when you are not dealing with gender  or LGTBI+ issues. Think in a blind person who is reading the paper will listen "Latino Slash A Slah X "we have to try to be inclusive also in our writing. You should write Latino and clarify in the  introduction that you are using these terms inclusively to encompass all genders.

---- > INTRODUCTION ;

2. When mentioning the number of incident cancers, please include the incidence rate between brakes that is very easy to compute (number of cases of Cancer/ population of the United States x 100 000).

3.       There is inconsistency in the citation methods used throughout the text. In some instances, you employ the Harvard referencing style, while in others, you use numeric citations. For coherence and to maintain a high academic writing standard, please use a single citation style consistently throughout your manuscript.

Here are a few examples from your text where the citation styles differ:

n  For those 27 to 45 years old who were not vaccinated at a younger age, the vaccine might be recommended based on shared decision-making.[6]

n  Rahman et al. (2013) showed that women in the South presented with the lowest vaccination rates in the U.S..[9]

                     Please revise your manuscript to adhere to one citation style

4.       Please introduce references when you do the following statement "Although the literature contains numerous examples of studies that focused on 130 intervention efforts targeting pediatric and parent populations"

---à  METHODS

5.       In line 148 somethin is missing "include in our survey [insert citation]. Our study team

6.       Please include as an annex the questionnaire used.

7.       Provide measures of the questionnaire's validity to ensure the reliability and credibility of your findings. Specifically, please include the Cronbach's alpha coefficients foreach of the theoretical constructs measured in your survey.

8.        This point is not compulsory; you may or may not do it. You may also want to provide other relevant validity measures, such as  Construct Validity (factor analysis results)  other measures as splithalf

9.       Indicate which statistical software you used eg R, IBMSPSS, etc.

--à RESULTS

10.    Including all variable values in the text makes it difficult to read and follow. Instead of listing all the values for each variable within the text, please refer to the corresponding tables where this information is already presented

11.    . In Tables 3, 4 delete  the Beta and SE columns. They are redundant, with the OR and 95% confidence intervals being sufficient. e(Beta) is the OR and e(beta'/-1.96 SE) is the 95% confidence interval of the OR. The information is being provided twice. The only thing this does is make the table variegated and difficult to understand. 1.9

12.    In tables authors use “Hispanic or Latino/a/x  in general Hispanic r efers to people who are from, or have ancestry from, Spanish-speaking countries. This includes Spain and most of Latin America (except Brazil, where Portuguese is spoken) by the other hand Latino  refers to Latin American countries, regardless of the language spoken. This includes countries in Central and South America, Mexico, and the Caribbean. So in my opinion  hispanic is a sub-group of Latino. (The only exception is when we are using U.S. Census data because U.S. Census data uses  the terms "Hispanic," "Latino," and "Spanish"  as a unique category to describe ethnicity. Specifically, the Census Bureau says that  Hispanic, Latino, or Spanish origin refers to people who identify their origin as Mexican, Puerto Rican, Cuban, Central or South American, or other Spanish culture or origin regardless of race..)

----à   DISCUSSION

It is correct I don see any error

---à BIBLIOGRAPHY

13.    The reference number 1 )  Disease Control and Prevention. HPV and Oropharyngeal Cancer. 2021 [cited 2024 March 12]; Available from: https://www.cdc.gov/cancer/hpv/basic_info/hpv_oropharyngeal.htm   is not fully in compliance with the ACS (American Chemical Society) style guidelines

Here is the corrected reference in ACS style:

Center for Disease Control and Prevention. HPV and Oropharyngeal Cancer. Centers for Disease Control and Prevention. https://www.cdc.gov/cancer/hpv/basic_info/hpv_oropharyngeal.htm (accessed Mar 12, 2024).

Please review the bibliography and correct the same error.

Author Response

  1. Avoid the term "Latino/a/x individuals" is difficult to understand, and not adequate for an academic journal when you are not dealing with gender  or LGTBI+ issues. Think in a blind person who is reading the paper will listen "Latino Slash A Slah X "we have to try to be inclusive also in our writing. You should write Latino and clarify in the  introduction that you are using these terms inclusively to encompass all genders.

Response: We appreciate the reviewer’s suggestion regarding gender inclusivity and accessibility. However, after reconsideration, we have chosen to use the term Latine, as it provides better accessibility than Latino/a/x; is gender neutral or encompassing of male, female, and gender-expansive people; and is preferred by Spanish speakers over other options (e.g., Latinx or Latin@) due to its adherence to existing Spanish pronunciation. We have made this clarification in our Methods section.

INTRODUCTION ;

  1. When mentioning the number of incident cancers, please include the incidence rate between brakes that is very easy to compute (number of cases of Cancer/ population of the United States x 100 000).

Response: We have corrected this in the Introduction.

  1. There is inconsistency in the citation methods used throughout the text. In some instances, you employ the Harvard referencing style, while in others, you use numeric citations. For coherence and to maintain a high academic writing standard, please use a single citation style consistently throughout your manuscript.

Here are a few examples from your text where the citation styles differ:

n  For those 27 to 45 years old who were not vaccinated at a younger age, the vaccine might be recommended based on shared decision-making.[6]

n  Rahman et al. (2013) showed that women in the South presented with the lowest vaccination rates in the U.S..[9]

                     Please revise your manuscript to adhere to one citation style

Response: Thank you for catching these mistakes. We have corrected our citations and now adhere to the journal’s style guide.

  1. Please introduce references when you do the following statement "Although the literature contains numerous examples of studies that focused on 130 intervention efforts targeting pediatric and parent populations".

Response: Thank you for this suggestion. We have provided several citations for systematic reviews that address these populations.

METHODS

  1. In line 148 somethin is missing "include in our survey [insert citation]. Our study team…

Response: Thank you for catching this oversight. We have corrected our error by deleting this sentence as the theory-driven constructs are not a focus of this analysis.

  1. Please include as an annex the questionnaire used.

Response: We fully intend to make the questionnaire available in full. However, as we are still finalizing the study and other related analyses, we reserve the right to make that available at a future date.

  1. Provide measures of the questionnaire's validity to ensure the reliability and credibility of your findings. Specifically, please include the Cronbach's alpha coefficients foreach of the theoretical constructs measured in your survey.
  2. This point is not compulsory; you may or may not do it. You may also want to provide other relevant validity measures, such as  Construct Validity (factor analysis results)  other measures as splithalf

Response: Thank you for these suggestions (item 7 and 8). We will provide this information in a future analysis that considers the constructs used in our survey. As this study is focused on demographic and socioeconomic predictors and does not include any of the theoretical constructs, we did not include that in this manuscript. That work is on-going, and therefore, not included here.

  1. Indicate which statistical software you used eg R, IBMSPSS, etc.

Response: We have provided this information at the end of the Methods.

RESULTS

  1. Including all variable values in the text makes it difficult to read and follow. Instead of listing all the values for each variable within the text, please refer to the corresponding tables where this information is already presented

Response: We assume that the reviewer is referring to the Methods section where we describe in detail the different levels of the variables. We have removed all description of categorical variables from the text that are sufficiently described in the tables, as the reviewer suggests, except where we feel additional detail was warranted.

  1. In Tables 3, 4 delete  the Beta and SE columns. They are redundant, with the OR and 95% confidence intervals being sufficient. e(Beta) is the OR and e(beta'/-1.96 SE) is the 95% confidence interval of the OR. The information is being provided twice. The only thing this does is make the table variegated and difficult to understand. 1.9

Response: We agree with the reviewer and have deleted these columns. We have also deleted the p-value column, as that is also redundant.

  1. In tables authors use “Hispanic or Latino/a/x  in general Hispanic r efers to people who are from, or have ancestry from, Spanish-speaking countries. This includes Spain and most of Latin America (except Brazil, where Portuguese is spoken) by the other hand Latino  refers to Latin American countries, regardless of the language spoken. This includes countries in Central and South America, Mexico, and the Caribbean. So in my opinion  hispanic is a sub-group of Latino. (The only exception is when we are using U.S. Census data because U.S. Census data uses  the terms "Hispanic," "Latino," and "Spanish"  as a unique category to describe ethnicity. Specifically, the Census Bureau says that  Hispanic, Latino, or Spanish origin refers to people who identify their origin as Mexican, Puerto Rican, Cuban, Central or South American, or other Spanish culture or origin regardless of race..)

Response: We appreciate the reviewer’s explanation of these terms. However, we prefer to use both terms in our manuscript for two reasons. First, this is the way that this was asked on our survey, so we feel we should retain the language as much as possible. Second, there are clear preferences among people of Hispanic or Latino/a/x – or alternatively Latine – origins due to how they identify with their heritage, culture, and ancestry.

BIBLIOGRAPHY

  1. The reference number 1 )  Disease Control and Prevention. HPV and Oropharyngeal Cancer. 2021 [cited 2024 March 12]; Available from: https://www.cdc.gov/cancer/hpv/basic_info/hpv_oropharyngeal.htm   is not fully in compliance with the ACS (American Chemical Society) style guidelines

Here is the corrected reference in ACS style:

Center for Disease Control and Prevention. HPV and Oropharyngeal Cancer. Centers for Disease Control and Prevention.  https://www.cdc.gov/cancer/hpv/basic_info/hpv_oropharyngeal.htm (accessed Mar 12, 2024).

Please review the bibliography and correct the same error.

Response: Thank you for catching these issues. We have addressed the specific issue identified and used the EndNote style guide to address any remaining issues.

Reviewer 3 Report

Comments and Suggestions for Authors

Cernasev et al. have conducted a significant cross-sectional survey to examine the factors influencing HPV vaccination uptake among Tennessee residents aged 18 to 45 years. Given the context of HPV being the most prevalent sexually transmitted infection in the U.S., and the concerning low vaccination rates in Tennessee—where over a thousand cancers were linked to HPV in 2020—the study's relevance is underscored. The survey, which accrued 2,011 complete responses, found substantial demographic differences in vaccination rates. Key findings indicate that females, younger adults, individuals with higher education, Hispanic or Latino/a/x respondents, and those with moderate to liberal political views are more likely to be vaccinated. Employment status also emerged as a critical predictor. Based on these insights, the authors recommend that public health interventions in Tennessee need to be tailored to these demographic characteristics to effectively enhance HPV vaccination coverage and diminish cancer risks associated with the virus.

The claims are properly placed in the context of the previous literature. The experimental data support the claims. The manuscript is written clearly enough that most of it is understandable to non-specialists. The authors have provided adequate proof for their claims, without overselling them. The authors have treated the previous literature fairly. The paper offers enough details of methodology so that the experiments could be reproduced.

Comments

1. In the introduction, the authors state, 'Before the approval of the first HPV vaccine, the healthcare team relied on screening as a preventive measure to combat these cancers.' It's important to note, however, that national screening programs for HPV-related cancers have been predominantly limited to cervical cancer. While there has been some screening for anal cancer among HIV-positive men who have sex with men (MSM), this is generally restricted to a few areas in a limited number of countries. Clarifying this could help refine the accuracy of the discussion on historical prevention strategies for HPV-related cancers.

2. In the introduction, the authors mention, 'The HPV vaccine is readily available and effective at preventing HPV-associated cancers; however, coverage and completion rates have been historically low despite being included as a Healthy People 2030 objective of 80% coverage of males and females ages 13–15. Thus, championing efforts to enhance routine vaccination could prevent most (92%) HPV-related cancers.' While it is accurate that the nonavalent HPV vaccine (Gardasil 9) could prevent up to 92% of all HPV-related cancer cases if coverage reaches 100% for both boys and girls, achieving only 80% coverage likely means a less than 92% reduction in these cancers across the population. Furthermore, realizing the full potential of the HPV vaccine in terms of cancer prevention will take considerable time—possibly 70-80 years—as it requires comprehensive vaccination across all age groups, not just those aged 13-15.

3. In the introduction, the authors cite a study by Boersma et al., noting that 'vaccination rates for one or more doses of the HPV vaccine in adults 18–26 years old increased from 22.1% to 39.9% from 2013 to 2028.' However, there appears to be a mistake in the reference period. According to reference [12], the study by Boersma et al. actually covers the years 2013 to 2018, not 2013 to 2028 as mentioned. It would be important to correct this to avoid confusion regarding the data timeline.

4. In the methods section, the authors state, 'We previously conducted a scoping review to identify relevant health behavior frameworks and related validated items and determine theoretical constructs to include in our survey [insert citation].' It appears that the citation intended to substantiate this claim is missing. Could the authors please provide the appropriate reference to ensure the readers can verify and understand the background of the methodologies used?

Minor revisions

Line 1-3, Title, "Demographic Influences on Adult HPV Vaccination - results from a cross-sectional survey in Tennessee"

Line 36-57, "Abstract: HPV remains the most prevalent sexually transmitted infection in the U.S., with over 80% of Americans contracting it by age 45. Despite the availability of effective vaccines, approved recently for adults aged 27-45, vaccination uptake is low across all age groups, particularly in Tennessee where 1,089 cancers were attributed to HPV in 2020. To understand the underlying factors, we conducted a cross-sectional survey from June 29 to August 17, 2023, among adults aged 18 to 45 in Tennessee. Using theory-based instruments, our survey explored vaccine-related beliefs, attitudes, and intentions. Analyzing responses from 2,011 participants via descriptive statistics, chi-square tests, and logistic regression models, we found higher vaccination rates among females, individuals aged 18-38, those with at least a high school education, Hispanic or Latino/a/x individuals, and those identifying as moderate or liberal. Notably, education and political views significantly influenced vaccination status. These insights underscore the need for public health interventions that consider these demographic differences to effectively increase vaccination rates and reduce HPV-associated cancer risks in Tennessee."

Line 62-64, "According to the Centers for Disease Control and Prevention (CDC), it is associated with approximately 46,711 new cancer diagnoses annually in the United States."

Line 64-66, "Cervical cancer, with 25,689 reported cases among women, and oropharyngeal cancer, with 21,022 cases among men, are the most common types of cancers caused by HPV."

Line 101-102, "However, the situation is more critical in the Southern states where vaccination rates are significantly lower compared to the North."

Line 301-308, "Our findings reveal that among adult residents in Tennessee, HPV vaccination rates are notably higher among females, individuals aged 18–38, those with at least some college education, moderates and liberals, Hispanic or Latino/a/x individuals, and employed respondents. To enhance vaccination rates and reduce the risk of HPV-associated cancers effectively, public health interventions must carefully consider these demographic differences. Future research should explore how these characteristics influence the facilitators and barriers to HPV vaccination."

Comments on the Quality of English Language

The manuscript is generally well-written. As a non-native English speaker, I have suggested some modifications to enhance clarity and fluency. However, I recommend professional copy-editing to further improve sentence construction, readability, coherence, and accuracy, ensuring a high-quality final presentation.

Author Response

  1. In the introduction, the authors state, 'Before the approval of the first HPV vaccine, the healthcare team relied on screening as a preventive measure to combat these cancers.' It's important to note, however, that national screening programs for HPV-related cancers have been predominantly limited to cervical cancer. While there has been some screening for anal cancer among HIV-positive men who have sex with men (MSM), this is generally restricted to a few areas in a limited number of countries. Clarifying this could help refine the accuracy of the discussion on historical prevention strategies for HPV-related cancers.

Response: Thank you for this valuable suggestion that helped us to clarify the statement.

  1. In the introduction, the authors mention, 'The HPV vaccine is readily available and effective at preventing HPV-associated cancers; however, coverage and completion rates have been historically low despite being included as a Healthy People 2030 objective of 80% coverage of males and females ages 13–15. Thus, championing efforts to enhance routine vaccination could prevent most (92%) HPV-related cancers.' While it is accurate that the nonavalent HPV vaccine (Gardasil 9) could prevent up to 92% of all HPV-related cancer cases if coverage reaches 100% for both boys and girls, achieving only 80% coverage likely means a less than 92% reduction in these cancers across the population. Furthermore, realizing the full potential of the HPV vaccine in terms of cancer prevention will take considerable time—possibly 70-80 years—as it requires comprehensive vaccination across all age groups, not just those aged 13-15.

Response: We appreciate the reviewer's comments on vaccination coverage and prevention of HPV-associated cancers. We understand that this is a nuanced situation and have tried to provide some additional clarification in the text to address this.

  1. In the introduction, the authors cite a study by Boersma et al., noting that 'vaccination rates for one or more doses of the HPV vaccine in adults 18–26 years old increased from 22.1% to 39.9% from 2013 to 2028.' However, there appears to be a mistake in the reference period. According to reference [12], the study by Boersma et al. actually covers the years 2013 to 2018, not 2013 to 2028 as mentioned. It would be important to correct this to avoid confusion regarding the data timeline.

Response: Thank you for this clarification. There was a typo, which was addressed.

  1. In the methods section, the authors state, 'We previously conducted a scoping review to identify relevant health behavior frameworks and related validated items and determine theoretical constructs to include in our survey [insert citation].' It appears that the citation intended to substantiate this claim is missing. Could the authors please provide the appropriate reference to ensure the readers can verify and understand the background of the methodologies used?

Response: Thank you for catching this omission. This has been corrected by deleting the statement as it is not directly relevant to this analysis.

Minor revisions

Line 1-3, Title, " Demographic Influences on Adult HPV Vaccination - results from a cross-sectional survey in Tennessee"

Response: Thank you for suggesting this new title, which is stronger.

Line 36-57, "Abstract: HPV remains the most prevalent sexually transmitted infection in the U.S., with over 80% of Americans contracting it by age 45. Despite the availability of effective vaccines, approved recently for adults aged 27-45, vaccination uptake is low across all age groups, particularly in Tennessee where 1,089 cancers were attributed to HPV in 2020. To understand the underlying factors, we conducted a cross-sectional survey from June 29 to August 17, 2023, among adults aged 18 to 45 in Tennessee. Using theory-based instruments, our survey explored vaccine-related beliefs, attitudes, and intentions. Analyzing responses from 2,011 participants via descriptive statistics, chi-square tests, and logistic regression models, we found higher vaccination rates among females, individuals aged 18-38, those with at least a high school education, Hispanic or Latino/a/x individuals, and those identifying as moderate or liberal. Notably, education and political views significantly influenced vaccination status. These insights underscore the need for public health interventions that consider these demographic differences to effectively increase vaccination rates and reduce HPV-associated cancer risks in Tennessee."

Response: We value your suggestions, and we have incorporated many of these into the revised abstract.

Line 62-64, "According to the Centers for Disease Control and Prevention (CDC), it is associated with approximately 46,711 new cancer diagnoses annually in the United States."

Response: We value your suggestion. The text was amended.

Line 64-66, "Cervical cancer, with 25,689 reported cases among women, and oropharyngeal cancer, with 21,022 cases among men, are the most common types of cancers caused by HPV."

Response: We value your recommendation. The text was amended.

Round 2

Reviewer 2 Report

Comments and Suggestions for Authors

The manuscript has incorporated all the considerations that were requested, in my opinion it is suitable for publication.